# The Cost of Replicability in Active Learning

**Rupkatha Hira**                                                    *rhira1@jh.edu*
*Department of Computer Science*
*Johns Hopkins University*
*A substantial portion of the work for this project was completed while the author was at the*
*University of Pennsylvania*

**Dominik Kau**                                                    *dominikk@upenn.edu*
*Penn Institute for Computational Science*
*University of Pennsylvania*

**Jessica Sorrell**                                                    *jess@jhu.edu*
*Department of Computer Science*
*Johns Hopkins University*
*A substantial portion of the work for this project was completed while the author was at the*
*University of Pennsylvania*

**Reviewed on OpenReview:** *https://openreview.net/forum?id=ZsqJu9eITd*

## Abstract

Active learning aims to reduce the number of labeled data points required by machine learning algorithms by selectively querying labels from initially unlabeled data. Ensuring replicability, where an algorithm produces consistent outcomes across different runs, is essential for the reliability of machine learning models but often increases sample complexity. This report investigates the cost of replicability in active learning using two classical disagreement-based methods: the CAL and $A^2$ algorithms. Leveraging randomized thresholding techniques, we propose two replicable active learning algorithms: one for realizable learning of finite hypothesis classes, and another for agnostic. Our theoretical analysis shows that while enforcing replicability increases label complexity, CAL and $A^2$ still achieve substantial label savings under this constraint. These findings provide key insights into balancing efficiency and stability in active learning.

## 1 Introduction

Modern machine learning techniques have demonstrated an impressive ability to improve model performance by training on increasing amounts of data. While unlabeled training data is abundant for many applications (e. g., text and image data sourced from the internet), obtaining large quantities of labeled data, required for classification and prediction tasks, can be prohibitively costly. For example, accurately labeling diagnostic imaging data requires medical expertise, so curating datasets for training medical risk predictors requires a great deal of clinician time and effort.

In response to these challenges, active learning has emerged as a powerful approach to reduce the number of labeled samples required to learn a good model (Angluin, 1988; Cohn et al., 1994b). Active learning algorithms selectively query the labels (or predicates of the labels) of data points that are most informative, while also leveraging unlabeled data to learn. A key challenge in active learning, as with any learning framework, is ensuring the stability of results, which is crucial for the robustness and reliability of machine learning models. Stability,

in the context of machine learning, refers to the insensitivity of an algorithm to perturbations in its training data. Informally, stability ensures that models do not overfit their training data, and a variety of stability notions have been studied for the purposes of guaranteeing generalization to unseen data and privacy-preservation of training data (Bousquet and Elisseeff, 2002; Dwork et al., 2006; Shalev-Shwartz et al., 2010; Dwork et al., 2015; Bassily et al., 2016).

In this work we consider the strong stability notion of *replicability*, introduced in Impagliazzo et al. (2022). Replicability requires that running a learning algorithm twice, on two independent datasets drawn from the same distribution and with shared internal randomness across both runs, yields identical models with high probability (over the samples and internal randomness). Replicable learning algorithms not only generalize well under adaptive data analysis (Impagliazzo et al., 2022), but they also enable verification of experiments in machine learning. By publishing the randomness used to train a model, another team of researchers can obtain the same model, using their own data, removing ambiguity in whether or not a replication effort has been successful. These properties come at the cost of increased sample complexity, however. In the case of PAC learning, e. g., it is known that the sample complexity of replicable learners depends on Littlestone dimension, as opposed to VC dimension as in non-replicable PAC-learning (Ghazi et al., 2021; Bun et al., 2023).

In this work, we investigate whether techniques from active learning can be employed to reduce the sample complexity overhead of replicable learning. We develop the first replicable algorithms in the active learning setting, giving realizable and agnostic learning algorithms for finite hypothesis classes. We prove that, indeed, there are natural conditions on the target distribution and hypothesis class under which our algorithms enjoy sample complexity improvements over passive learning, establishing the utility of active label queries in replicable learning.

## 1.1 Our Results

We give the first replicable algorithms for active learning of finite hypothesis classes, in both the realizable and agnostic setting. The sample complexity bounds for our replicable algorithms show an improvement in sample complexity over passive learning analogous to known improvements from active learning for non-replicable algorithms. More precisely, for target error rate $\varepsilon$, the sample bounds for our realizable algorithm has only logarithmic dependence on $\frac{1}{\varepsilon}$. For replicable passive learning, this dependence is linear, and so this represents a significant improvement in accuracy dependence. In the agnostic setting, our replicable active learning algorithm instead has a sample complexity dependence on $\left(\frac{\nu}{\varepsilon}\right)^2$, where $\nu$ is the error of the optimal hypothesis in the class $C$. While the dependence on $\frac{1}{\varepsilon}$ is technically still quadratic for our algorithm, as it is for the replicable passive agnostic learning algorithm of Bun et al. (2023), we note that when the optimal error $\nu$ is quite close to the target error $\varepsilon$, this still represents a significant improvement in accuracy dependence, and therefore we do still improve over passive replicable learning in both realizable and agnostic cases.

Similarly to Cohn et al. (1994a); Balcan et al. (2006); Hanneke (2007), we instead obtain a sample complexity dependence on the *disagreement coefficient* $\Theta$ of a hypothesis class $C$ for distribution $D$. We formally define the disagreement coefficient in section 2, but informally, the disagreement coefficient is a measure of the probability of disagreement among hypotheses in a class $C$ that are within some error ball centered on the optimal hypothesis in $C$. A small disagreement coefficient means that relatively few labeled samples are needed to rule out hypotheses that are far from optimal, with the caveat that these samples should be points on which hypotheses in $C$ disagree. Active learning allows us to selectively query such points, and therefore obtain sample complexities dependent on $\Theta$ and only logarithmically on $\frac{1}{\varepsilon}$ (or quadratically on $\frac{\nu}{\varepsilon}$ in the agnostic case). The sample complexity dependence on $\Theta$ we obtain for our replicable active learning algorithms is analogous to those in Cohn et al. (1994a);

Balcan et al. (2006); Hanneke (2007): linear dependence in the realizable case, and quadratic in the agnostic.

## 1.2 Related Work

Our replicable realizable PAC learner adapts techniques for replicable learning of finite hypothesis classes developed in Bun et al. (2023) to the CAL algorithm given in Cohn et al. (1994a). The CAL algorithm was analyzed and extended in Balcan (2015); Dasgupta et al. (2007); Hanneke et al. (2014) (see section 2.1.2 for a description of the CAL algorithm). Our replicable agnostic PAC learner builds on the work of Balcan et al. (2006); Dasgupta et al. (2007), taking the $A^2$ algorithm as a starting point for our algorithm. The $A^2$ algorithm was the first active learning algorithm to achieve $\varepsilon$-optimal performance where the underlying distribution has arbitrary noise (see section 2.1.3 for a description of the $A^2$ algorithm).

Castro and Nowak (2008); Dasgupta (2004) study the limits on the sample complexity improvements achievable by active learning. In particular Dasgupta (2004) show that even for the very simple hypothesis class of $d$-dimensional linear separators, there are target hypotheses for which active label queries cannot provide significant sample complexity improvements over passive learning. These fundamental limits motivated a line of work studying more expressive queries, such as comparison queries, which enabled exponential sample complexity improvements over label query active learning algorithms in some cases (Kane et al., 2017; Hopkins et al., 2020a;b). To initiate the study of replicability in active learning, we restrict our algorithms to make only label queries. Thus, our sample complexity improvements will depend on the disagreement coefficient $\Theta$ of the hypothesis class $C$ and distribution $D$, and will not be guaranteed to hold for arbitrary distributions and classes.

Prior work has studied active learning under the related stability constraint of differential privacy (Balcan and Feldman (2013), Bittner et al. (2020), Ghassemi et al. (2016)). The connection between privacy and replicability was studied in Ghazi et al. (2021); Kalavasis et al. (2023); Bun et al. (2023), but the equivalence between the two was established only for statistical tasks in the batch setting. Hence, it is not immediately clear how to leverage this equivalence to obtain replicable learning algorithms from private ones in the active learning context. Replicable algorithms have been developed for other learning models outside of the batch PAC learning framework as well. Prior work has given replicable algorithms for sequential decision-making problems such as bandits (Esfandiari et al., 2022), online learning (Ahmadi et al., 2024), and reinforcement learning (Karbasi et al., 2023; Eaton et al., 2024), but none had yet been given for active learning.

## 1.3 Organization

The structure of this paper is as follows. In section 2, we introduce the theory of active learning, followed by the concept of replicability in machine learning. In section 3, we propose an algorithm for replicable active learning in the realizable setting and analyze its convergence. In section 4, we adapt the algorithm of section 3 to the agnostic setting and provide an analysis. Finally, we conclude with suggestions for future work. The appendix includes complete proofs and appendix A listing symbols used throughout the paper.

## 2 Background

### 2.1 Active Learning Theory

We work in the PAC learning framework of Valiant (1984). Fix a domain $X$, a binary label space $Y = \{0, 1\}$, a concept class $C$ of hypotheses $h : X \to Y$ and define the ground truth or target function as $c \in C$. Algorithm $\mathcal{A}$ is said to be a PAC learner for $C$ if there exists a function $m(\varepsilon, \delta)$, polynomial in $\frac{1}{\varepsilon}, \frac{1}{\delta}$, such that for every distribution $D$ over $X$ and every $\varepsilon$ and $\delta > 0$, given $m(\varepsilon, \delta)$ samples $x \in X$ drawn i.i.d. from $D$ and labels $y = c(x) \in Y$, $\mathcal{A}$ outputs a hypothesis $h \in C$ such that $\mathrm{err}_D(h) = \Pr_{x \sim D}[h(x) \neq y] < \varepsilon$, except with probability $\delta$ over the choice of samples.

In the agnostic setting, the ground truth is not a hypothesis in $C$, and therefore the minimum error achievable by a hypothesis $h \in C$ may be non-zero. We will use $\nu$ — sometimes called the noise rate — to denote the error of the optimal hypothesis in $C$: $h^* = \operatorname{argmin}_{h \in C} \operatorname{err}(h)$. An agnostic learning algorithm $\mathcal{A}$ will return a hypothesis $h$ with error that does not exceed the error rate $\nu$ by more than $\varepsilon$.

In the active learning framework, learning algorithms do not receive labels for all samples from $D$. Instead, they are assumed to have access to essentially unlimited unlabeled data, and their goal is to learn the ground truth function $h$ by making targeted queries for labels (or functions of the labels). The aim of active learning is to improve the sample — and especially label — complexity of learning relative to passive algorithms for the equivalent task. This is especially useful for tasks where the unlabeled sample points are easily accessible but the labeling requires additional (e. g. computational or manual) effort. Examples of such tasks include image classification and speech recognition.

The field of active learning can be further subdivided based on how queries are contrived and how points are sampled. Algorithms that select queries via the query-by-disagreement principle base their queries on the disagreement of all candidate hypotheses. Stream-based selective sampling encompasses algorithms that receive one sample point at a time and determine for each point if they want to request a label or not (Settles, 2012).

### 2.1.1 Disagreement Coefficient

The set of $x \in X$ on which at least two hypotheses $h$ from a version space $V \subseteq C$ disagree is defined as

$$DIS(V) = \{x \in X \mid \exists h_1, h_2 \in V \text{ s.t. } h_1(x) \neq h_2(x)\}. \tag{1}$$

In the following, this set is referred to as the disagreement region or set. The respective probability of sampling an $x$ in the disagreement region is

$$\Delta_D(V) = \Pr_{x \sim D}[x \in DIS(V)] \tag{2}$$

with probability distribution $D$. The distance metric for two hypotheses $h_1, h_2$

$$d_D(h_1, h_2) = \Pr_{x \sim D}[h_1(x) \neq h_2(x)] \tag{3}$$

is used to define a ball around a hypothesis

$$B_D(h, \varepsilon) = \{h' \in C \mid d_D(h, h') \leq \varepsilon\}. \tag{4}$$

The ball $B_D(c, \varepsilon)$ where $c$ is the target function includes all hypotheses with an error rate of at most $\varepsilon$. Then, $\Delta_D(B_D(c, \varepsilon))$ is the probability of sampling a point from distribution $D$ on which at least two hypotheses with an error rate of at most $\varepsilon$ disagree. The disagreement coefficient is defined as

$$\Theta_D = \sup_{\varepsilon > 0} \frac{\Delta_D(B_D(c, \varepsilon))}{\varepsilon} \tag{5}$$

and describes the maximum aforementioned probability normalized by $\varepsilon$. Intuitively, this is a measure of how many points have to be sampled to improve upon a set of hypotheses with an error rate of at most $\varepsilon$. In the agnostic case the definition uses $h^*$ instead of $c$.

This becomes clear when considering the worst case round of the CAL algorithm, which will be explained in the next section. It is clear that the worst case occurs when all points in the current disagreement region have to be sampled to remove all hypotheses with an error rate greater than $\varepsilon$. Thus, consider the case where the target function and $n$ additional hypotheses remain in the version space. Each of the $n$ hypotheses makes a mistake only on a single point that is sampled with a probability of $\frac{1}{n}$. Then, the disagreement region has a probability mass of $n \cdot \frac{1}{n} = 1$, and the disagreement coefficient for the critical error rate $\frac{1}{n}$ is $n$ — the number of points that have to be sampled.

In the following the subscript $D$ will be omitted from the introduced variables for succinctness if the respective probability distribution is clear from context.

### 2.1.2 CAL Algorithm

The CAL algorithm, which was first proposed by and named after Cohn et al. (1994a) is based on the concept of query by disagreement and is used for learning in the realizable case. The algorithm is given in algorithm 1 as pseudo-code. Despite the pooling of points in each round $r$, the algorithm is categorized as a stream-based selective sampling algorithm. The choice over requesting a label depends on whether a given point is in the disagreement region. This is equivalent to sampling from an alternate probability distribution $D_r$ that is obtained by conditioning on the inclusion in the disagreement region. In this round-based formulation the probability mass of the disagreement region is at least halved in each round. The exit condition of the loop ensures that all hypotheses in the final version space will have an error rate smaller than $\varepsilon$. This results from the fact that the target function $c$ is never eliminated and no hypothesis may deviate more than $\varepsilon$ from the ground truth. Furthermore, it follows that the number of rounds is $\mathcal{O}\left(\log \frac{1}{\varepsilon}\right)$.

A detailed label complexity analysis of such a disagreement region based algorithm in terms of the disagreement coefficient $\Theta$ was first derived in Balcan et al. (2006). The label complexity for a finite hypothesis class as given by Hsu (2010) is

$$
\mathcal{O}\left(\log \frac{1}{\varepsilon} \cdot \Theta \log \frac{|C| \log \frac{1}{\varepsilon}}{\delta}\right). \tag{6}
$$

Here, the first factor accounts for the number of rounds that the CAL algorithm will run for and the second factor is the number of points $k$ that are sampled in each round. Compared to the sample complexity of a passive learner Kearns and Vazirani (1994)

$$
\mathcal{O}\left(\frac{1}{\varepsilon} \log \frac{|C|}{\delta}\right), \tag{7}
$$

the CAL algorithm yields an exponential improvement in label complexity with respect to the dependence on $\varepsilon$, assuming that the disagreement coefficient is finite.

---

**Algorithm 1** CAL algorithm

---

**input:** $\delta, \varepsilon$
1: Set sample size $k = \mathcal{O}\left(\Theta \log \frac{|C| \log \frac{1}{\varepsilon}}{\delta}\right)$
2: Initialize version space $V = C$
3: **while** $\Delta(V) > \varepsilon$ **do**
4:      Sample $k$ points $x_1, \ldots, x_k$ from $DIS(V)$
5:      Query labels $y_1, \ldots, y_k$ for sampled points
6:      Update $V \leftarrow \{h \in V : \forall i \in [k] : h(x_i) = y_i\}$
7: **end while**
8: **return** Any $h \in V$

---

### 2.1.3 A² Algorithm

The A² algorithm was first proposed by Balcan et al. (2006), as the first agnostic active learning algorithm. It can be thought of as a robust version of the CAL algorithm that allows for noise. It is a disagreement-based active learning algorithm that was shown to work in an agnostic setting with no assumptions about the mechanism producing noise. All it needs access to is a stream of examples drawn i.i.d from some fixed distribution.

The algorithm is given in algorithm 2 as pseudo-code. This pseudocode is chosen from Balcan (2015), over other flavors of the algorithm as depicted in Balcan et al. (2006) and ..., for the sake of simplicity.

To work in the Agnostic setting, the A² algorithm must be more conservative than the CAL algorithm. The rejection of bad hypotheses based on disagreement over a single example can

no longer be a valid step, since it would risk rejecting the best hypothesis with a non-zero noise rate. Instead, in each round it estimates the distributional lower and upper bounds, and eliminates all hypotheses from the disagreement region whose lower bound is greater than the minimum estimated upper bound. Similar to the CAL algorithm, the probability mass of the disagreement region is at least halved in each round. Since the exit condition of the loop is relatively weaker, the algorithm concludes with one last step where a certain number of points are all labeled and the hypothesis from the remaining version space which has the lowest estimated error is finally chosen. The error of the final hypothesis is provably smaller than $\nu + \varepsilon$ where $\nu$ is the noise rate, or the true error of the ground truth.

It follows from the exit condition that the number of rounds in the loop is $\mathcal{O}\left(\log \frac{1}{\Theta \nu}\right)$.

The label complexity for a finite hypothesis class as given by Hanneke (2007) is

$$\mathcal{O}\left(\Theta^2 \log \frac{1}{\Theta \nu} \left(\frac{\nu^2}{\varepsilon^2} + 1\right) \left(\log |C| + \log \frac{1}{\delta}\right)\right). \tag{8}$$

Compared to the sample complexity of a passive agnostic (PAC) learner in Kearns and Vazirani (1994)

$$\mathcal{O}\left(\frac{1}{\varepsilon^2} \left(\log |C| + \log \frac{1}{\delta}\right)\right), \tag{9}$$

the A$^2$ algorithm yields a significant improvement in label complexity with respect to the dependence on $\varepsilon$, assuming that the disagreement coefficient is finite, and the noise rate is small enough.

---

**Algorithm 2** $A^2$ algorithm

---

**input:** $\nu, \delta, \varepsilon$

1: Initialize $V_i = C$, $k = \tilde{\mathcal{O}}\left(\Theta^2 d\right)$, $k' = \tilde{\mathcal{O}}\left(\frac{\Theta^2 d \nu^2}{\varepsilon^2}\right)$, $\delta' = \frac{\delta}{1 + \lceil \log \frac{1}{8 \Theta \nu} \rceil}$.

2: **while** $\Delta(V_i) \geq 8\Theta\nu$ **do**
    (a) Let $D_i$ be the conditional distribution $D$ given that $x \in DIS(V_i)$.
    (b) Sample $k$ i.i.d labeled examples from $D_i$. Denote this set by $S_i$.
    (c) Update $V_{i+1} = \{h \in V_i : LB(S_i, h, \delta') \leq \min_{h' \in V_i} UB(S_i, h', \delta')\}$.

3: **end while**

4: Sample $S$ of $k'$ points from $D_i$.

5: **return** $\arg\min_{h \in V_i} \mathrm{err}_S(h)$

---

## 2.2 Replicability in Learning

The notion of replicability we use in our work was introduced by Impagliazzo et al. (2022), to define randomized learning algorithms that are stable with high probability over different samples from the same underlying distribution. Following is the definition of replicability introduced by Impagliazzo et al. (2022) that we adopt in our work.

A randomized algorithm $\mathcal{A}(S; b)$ is replicable if there exists a function $m_0 : \mathbb{R} \to \mathbb{N}$ such that for all $\rho > 0$, and any $m > m_0(\rho)$

$$\Pr_{S_1, S_2, b} [\mathcal{A}(S_1; b) = \mathcal{A}(S_2; b)] \geq 1 - \rho, \tag{10}$$

where $S_1$ and $S_2$ denote samples of size $m$ drawn i.i.d. from $D$, and $b$ denotes a random binary string representing the internal randomness used by $\mathcal{A}$. We will call learning algorithms that are simultaneously replicable and PAC learners *replicable learning algorithms*.

**Replicable Learner for Finite Classes**  To develop our efficient RepliCAL algorithm, we have drawn from the random thresholding trick used to develop a replicable learner for finite hypothesis classes in Bun et al. (2023). The idea is to estimate the risk of each

hypothesis in the class $C$ by standard uniform convergence bounds, choose a random error threshold $v \in [OPT, OPT + \alpha]$, and finally output a random $h \in C$ with empirical error $\text{err}_S(h) = \frac{1}{|S|} \sum_{(x,y) \in S} \mathbf{1}[h(x) \neq y]$ guaranteed to be at most $v$. It was shown in the paper that such random thresholding achieves replicability with high probability when the hypothesis class is finite.

In the realizable case, the required sample complexity for this learner was shown to be

$$\mathcal{O}\left(\frac{\log^2 |C| \log \frac{1}{\rho} + \rho^4 \log\left(\frac{1}{\delta}\right)}{\varepsilon \rho^4}\right) \tag{11}$$

This result was further improved upon with regards to the replicability parameter $\rho$ by a boosting procedure. Then, the resulting sample complexity for the realizable case is

$$\mathcal{O}\left(\log^3 \frac{1}{\rho} \cdot \frac{\log^2 |C| + \log\left(\frac{1}{\rho\beta}\right)}{\varepsilon \rho^2}\right) \tag{12}$$

In our work we have extended the random thresholding concept to the active learning setting and proved that it leads to replicable learning.

In the last section, we propose an agnostic replicable learner for finite classes, with a label complexity of

$$\tilde{\mathcal{O}}\left(\Theta^2 \left(\log \frac{1}{\Theta\nu} + \frac{\nu^2}{\varepsilon^2}\right) \cdot \left(\log \frac{|C|}{\delta} + \frac{\log^2 |C| \log \frac{1}{\rho} \log^4 \frac{1}{\Theta\nu}}{\rho^4}\right)\right). \tag{13}$$

The dependence on $\rho$ can be brought down by boosting, and the resulting label complexity would be

$$\tilde{\mathcal{O}}\left(\Theta^2 \log^3 \frac{1}{\rho} \left(\log \frac{1}{\Theta\nu} + \frac{\nu^2}{\varepsilon^2}\right) \cdot \left(\log \frac{|C|}{\rho\delta} + \frac{\log^2 |C| \log^4 \frac{1}{\Theta\nu}}{\rho^2}\right)\right). \tag{14}$$

## 3 Replicable Active Realizable Learning

### 3.1 Algorithm

Our approach is based on the replicable learning algorithm for finite hypothesis classes given by Bun et al. (2023). In each loop of the RepliCAL algorithm, the version space is updated by thresholding the empirical, conditional error rate based on a random threshold which is selected at the start. To compute the conditional error rate, the algorithm exclusively queries labels of points in the disagreement region. The size of the disagreement region is calculated using unlabeled data, and once it is smaller than the target error rate, the algorithm exits the loop. After exiting the loop, all hypotheses in the final version space will be randomly reordered, and the first hypothesis returned. Replicability is achieved by ensuring that for two different runs of the algorithm the final version spaces are similar and therefore the same hypothesis will be returned with high probability. Importantly, we do not require the per-round version spaces to be similar across independent runs; our analysis only couples the terminal version spaces.

### 3.2 Theoretical Analysis

**Theorem 1.** *Let $C$ be any finite concept class. In the realizable setting, RepliCAL is a replicable active learning algorithm for $C$ with label complexity:*

$$\mathcal{O}\left(\Theta \log \frac{1}{\varepsilon} \cdot \frac{\log^2 |C| \log \frac{\log \frac{1}{\varepsilon}}{\rho} \log^4 \frac{1}{\varepsilon} + \rho^4 \log \frac{|C| \log \frac{1}{\varepsilon}}{\delta}}{\rho^4}\right). \tag{15}$$

---

**Algorithm 3** RepliCAL algorithm

---

**input:** $\delta, \varepsilon, \rho$

1: Set interval size $\tau = \mathcal{O}\left(\frac{\rho^2}{\Theta \log |C|}\right)$

2: Set sample size $k = \tilde{\mathcal{O}}\left(\Theta \log \frac{1}{\varepsilon} \cdot \frac{\log^2 |C| \log \frac{1}{\rho} \log^4 \frac{1}{\varepsilon} + \rho^4 \log \frac{|C|}{\delta}}{\rho^4}\right)$

3: Initialize version space $V = C$

4: Select random threshold $v \leftarrow \left\{\frac{1}{2}\tau, \frac{3}{2}\tau, \ldots, \frac{1}{8\Theta} - \frac{\tau}{2}\right\}$

5: **while** $\Delta(V) \geq \varepsilon$ **do**

6:      Sample $k$ points $x_1, \ldots, x_k$ from $DIS(V)$

7:      Query labels $y_1, \ldots, y_k$ for sampled points

8:      Define set $S_r = \{(x_1, y_1), \ldots, (x_k, y_k)\}$

9:      Estimate conditional error $\mathrm{err}_{S_r}^{D_r}(h)$ for every $h \in V$

10:      $V \leftarrow \{h \in V : \mathrm{err}_{S_r}^{D_r}(h) \leq v\}$

11: **end while**

12: Sample $k$ points $x_1, \ldots, x_k$ from $DIS(V)$

13: Query labels $y_1, \ldots, y_k$ for sampled points

14: Define set $S_r = \{(x_1, y_1), \ldots, (x_k, y_k)\}$

15: Estimate conditional error $\mathrm{err}_{S_r}^{D_r}(h)$ for every $h \in V$

16: Set $v' = \frac{\varepsilon v}{\Delta(V)}$

17: $V \leftarrow \{h \in V : \mathrm{err}_{S_{R+1}}^{D_{R+1}}(h) \leq v'\}$

18: Randomly order all $h \in V$

19: **return** The first hypothesis in $V$

---

We prove theorem 1 via lemma 1 and lemma 2, which separately establish accuracy and replicability of algorithm 3.

**Lemma 1.** *Let $\varepsilon, \delta, \rho > 0$ respectively denote accuracy, failure, and replicability parameters. Let $m(\varepsilon, \delta, \rho, |C|)$ denote the total (labeled and unlabeled) sample complexity for algorithm 3. Then for any finite hypothesis class $C$ and distribution $D$, except with probability at most $\delta$ over $S \sim D^m$, RepliCAL terminates after $\mathcal{O}\left(\log \frac{1}{\varepsilon}\right)$ rounds and outputs a hypothesis $h$ with error at most $\varepsilon$.*

The proof follows closely that of Balcan (2015) and is given in detail in appendix B.1.

It remains to argue that algorithm 3 is replicable. We will follow the proof approach of Bun et al. (2023). Let $V^1$ and $V^2$ denote the final sets of candidate hypotheses upon exiting the main loop of RepliCAL, for two independent runs of the algorithm with resampled data, but shared internal randomness. We argue that the symmetric difference $V^1 \Delta V^2$ is small relative to their union $V^1 \cup V^2$, and therefore returning the first element of a random permutation of $C$ that is contained in $V^1$ (resp. $V^2$) returns the same hypothesis with high probability.

**Lemma 2.** *Let $\varepsilon, \delta, \rho > 0$ respectively denote accuracy, failure, and replicability parameters. Let $m(\varepsilon, \delta, \rho, |C|)$ denote the total (labeled and unlabeled) sample complexity for algorithm 3. Then for any finite hypothesis class $C$ and distribution $D$,*

$$\Pr_{\substack{S_1, S_2 \sim D^m \\ b}}[\mathrm{RepliCAL}(S_1; b) \neq \mathrm{RepliCAL}(S_2; b)] < \rho. \tag{16}$$

The complete proof is given in appendix B.2 and the proof of theorem 1 then follows as a corollary of lemma 1 and lemma 2, by an accounting of the labeled sample complexity as given in appendix B.3.

### 3.3 Boosting

The label complexity can be boosted via the procedure proposed in Impagliazzo et al. (2022) and modified in Bun et al. (2023) to improve the dependence on $\rho$. The boosting

procedure is based on the idea of running the replicable learning algorithm on $\mathcal{O}\left(\log\frac{1}{\rho}\right)$ different random strings with a constant replicability parameter $\rho' = 0.01$. Different sets of samples induce a distribution of hypotheses for each random string. Because of the constant replicability parameter, with high probability at least one of these distributions will have a $\Omega(1)$ heavy-hitter, i.e. an element that is drawn with extremely high probability. The rHeavyHitters algorithm given in Impagliazzo et al. (2022) is used to replicably find a heavy-hitter hypothesis for which it requires $\mathcal{O}\left(\frac{\log^3(1/\rho)}{\rho^2}\right)$ samples that are shared between the multiple runs on different random strings.

Setting the failure probability during the repeated running of the replicable learning algorithm to $\delta' = \delta \cdot \frac{\rho^2}{\log^3(1/\rho)} \approx \mathcal{O}\left(\delta \cdot \text{poly}(\rho)\right)$ ensures that — by a union bound over all samples — the hypotheses will be good with probability $1 - \delta$. Therefore, the $\log\frac{1}{\delta}$ term of the non-boosted version is changed to $\log\frac{1}{\rho\delta}$.

This results in a label complexity of

$$\mathcal{O}\left(\Theta\log\frac{1}{\varepsilon}\log^3\frac{1}{\rho} \cdot \frac{\log^2|C|\log\log\frac{1}{\varepsilon}\log^4\frac{1}{\varepsilon} + \log\frac{|C|\log\frac{1}{\varepsilon}}{\delta\rho}}{\rho^2}\right). \tag{17}$$

Analogous to equation 51, this can be approximated as

$$\tilde{\mathcal{O}}\left(\Theta\log\frac{1}{\varepsilon}\log^3\frac{1}{\rho} \cdot \frac{\log^2|C|\log^4\frac{1}{\varepsilon} + \log\frac{|C|}{\delta\rho}}{\rho^2}\right). \tag{18}$$

### 3.4 Comparison to Replicability in Passive Learning

A direct comparison of the label complexity we obtained in equation 51 to the passive replicable learning guarantee of Bun et al. (2023) in equation 11 reveals a clear improvement in sample complexity whenever the hypothesis class and data distribution admit efficient active learning, as captured by a bounded disagreement coefficient $\Theta$. In particular, while passive replicable learning must draw labeled examples from the full underlying distribution in order to ensure stability of the learned hypothesis, RepliCAL concentrates label queries only within the evolving disagreement region. This allows the algorithm to simultaneously shrink the version space and maintain replicability, while avoiding the need to repeatedly label regions on which all surviving hypotheses already agree. As a consequence, the resulting sample complexity exhibits only polylogarithmic dependence on $1/\varepsilon$, in contrast to the linear dependence in the realizable passive replicable learner of Bun et al. (2023). Thus, active learning not only reduces label complexity in the standard PAC sense, but also mitigates the additional sampling burden imposed by replicability constraints. Overall, this demonstrates that active learning provides a principled avenue for overcoming the high sample complexity traditionally associated with replicable learning.

## 4 Replicable Active Agnostic Learning

### 4.1 Algorithm

In this section, we introduce the ReplicA$^2$ algorithm (algorithm 4) for replicable active learning in the agnostic setting. Algorithm 4 adapts the approach of algorithm 3 to the agnostic setting, by removing the implicit assumption that there always exists a perfectly consistent hypothesis within the version space $V$. This requires determining the size of the disagreement region exactly not only to determine when to exit the main loop, but also to approximate an upper bound on the global error of the optimal hypothesis at each round using only labeled samples from the disagreement region, so that we can remove any hypothesis with conditional error exceeding this bound.

---

**Algorithm 4** ReplicA$^2$ algorithm

---

**input:** $\delta, \varepsilon, \rho, b$

1: Set interval size $\tau = \mathcal{O}\left(\frac{\rho^2}{\Theta \log |C|}\right)$

2: Set labeled sample size $k = \tilde{\mathcal{O}}\left(\Theta^2 \log \frac{1}{\Theta\nu}\left(\log \frac{|C|}{\delta} + \frac{\log^2 |C| \log \frac{1}{\rho} \log^4 \frac{1}{\Theta\nu}}{\rho^4}\right)\right)$

3: Set labeled sample size $k' = \tilde{\mathcal{O}}\left(\Theta^2 \frac{\nu^2}{\varepsilon^2}\left(\log \frac{|C|}{\delta} + \frac{\log^2 |C| \log \frac{1}{\rho} \log^4 \frac{1}{\Theta\nu}}{\rho^4}\right)\right)$

4: Initialize version space $V = C$

5: Select random threshold $v \leftarrow_b \left\{\frac{1}{16\Theta} + \frac{1}{2}\tau, \frac{1}{16\Theta} + \frac{3}{2}\tau, \ldots, \frac{3}{16\Theta} - \frac{\tau}{2}\right\}$

6: **while** $\Delta(V) \geq 8\Theta\nu$ **do**

7:     Define $\sigma_r = \frac{\nu}{\Delta(V)}$

8:     Sample $k$ points $x_1, \ldots, x_k$ from $DIS(V)$

9:     Query labels $y_1, \ldots, y_k$ for sampled points

10:     Define set $S_r = \{(x_1, y_1), \ldots, (x_k, y_k)\}$

11:     Estimate conditional error $\mathrm{err}_{S_r}^{D_r}(h)$ for every $h \in V$

12:     $V \leftarrow \left\{h \in V : \mathrm{err}_{S_r}^{D_r}(h) \leq v + \sigma_r\right\}$

13: **end while**

14: Set interval size $\tau' = \mathcal{O}\left(\frac{\varepsilon\rho^2}{\Delta(V) \log |C|}\right)$

15: Select threshold $v'$ in $\left\{\frac{\varepsilon}{96\Theta\nu} + \frac{\tau'}{2}, \frac{\varepsilon}{96\Theta\nu} + \frac{3}{2}\tau', \ldots, \frac{2\varepsilon}{96\Theta\nu} - \frac{\tau'}{2}\right\}$ with the same interval index as before

16: Sample $k'$ points $x_1, \ldots, x_{k'}$ from $DIS(V)$

17: Query labels $y_1, \ldots, y_{k'}$ for sampled points

18: Define set $S_{R+1} = \{(x_1, y_1), \ldots, (x_{k'}, y_{k'})\}$

19: Estimate conditional error $\mathrm{err}_{S_{R+1}}^{D_{R+1}}(h)$ for every $h \in V$

20: Define conditional optimal error as $\widehat{\nu}^{D_{R+1}} = \min_{h \in V} \mathrm{err}_{S_{R+1}}^{D_{R+1}}(h)$

21: $V \leftarrow \{h \in V : \mathrm{err}_{S_{R+1}}^{D_{R+1}}(h) \leq \widehat{\nu}^{D_{R+1}} + v'\}$

22: Randomly order all $h \in V$

23: **return** The first hypothesis in $V$

---

In the final round, $\nu$ scaled by the size of the disagreement region no longer provides a useful upper bound on the conditional optimal error, and so the algorithm instead takes the minimum conditional error as an estimate. Analogous to the realizable case, replicability is achieved by ensuring that the final version spaces of two different runs of the algorithm are similar, and that therefore the same hypothesis will be returned with high probability.

### 4.2 Theoretical Analysis

**Theorem 2.** *Let $C$ be any finite concept class. In the agnostic setting, ReplicA$^2$ is a replicable active learning algorithm for $C$ with label complexity:*

$$\tilde{\mathcal{O}}\left(\Theta^2\left(\log \frac{1}{\Theta\nu} + \frac{\nu^2}{\varepsilon^2}\right)\left(\log \frac{|C|}{\delta} + \frac{\log^2 |C| \log \frac{1}{\rho} \log^4 \frac{1}{\Theta\nu}}{\rho^4}\right)\right). \tag{19}$$

As with theorem 1, we prove theorem 2 in two lemmas separately arguing for accuracy and replicability. For brevity, the proofs of both lemmas are omitted here and presented in the appendix under appendix C.1 and appendix C.2.

**Lemma 3.** *Let $\varepsilon, \delta, \rho > 0$ respectively denote accuracy, failure, and replicability parameters. Let $m(\varepsilon, \delta, \rho, |C|)$ denote the total (labeled and unlabeled) sample complexity for algorithm 4. Then for any finite hypothesis class $C$ and distribution $D$, except with probability at most $\delta$ over $S \sim D^m$, ReplicA$^2$ terminates after $\mathcal{O}\left(\log \frac{1}{\Theta\nu}\right)$ rounds and outputs a hypothesis $h$ with error at most $\nu + \varepsilon$, where $\nu$ denotes the error of the optimal hypothesis in $C$.*

**Lemma 4.** *Let $\varepsilon, \delta, \rho > 0$ respectively denote accuracy, failure, and replicability parameters. Let $m(\varepsilon, \delta, \rho, |C|)$ denote the total (labeled and unlabeled) sample complexity for algorithm 4. Then for any finite hypothesis class $C$ and distribution $D$,*

$$\Pr_{\substack{S_1, S_2 \sim D^m \\ b}}[\text{ReplicA}^2(S_1; b) \neq \text{ReplicA}^2(S_2; b)] < \rho. \tag{20}$$

Similarly to the realizable case, theorem 2 follows as a corollary of lemma 3 and lemma 4. A detailed derivation is given in appendix C.3 Applying Boosting to the algorithms using the same setup as in section 3.3, we can reduce this complexity to

$$\mathcal{O}\left(\frac{\Theta^2 \log^3 \frac{1}{\rho}}{\rho^2}\left[\left(\log\frac{1}{\Theta\nu} + \frac{\nu^2}{\varepsilon^2}\right)\log^2|C|\log\log\frac{1}{\Theta\nu}\log^4\frac{1}{\Theta\nu}\right.\right.$$
$$\left.\left.+ \log\frac{1}{\Theta\nu}\log\frac{|C|\log\frac{1}{\Theta\nu}}{\rho\delta} + \frac{\nu^2}{\varepsilon^2}\log\frac{|C|}{\rho\delta}\right]\right) \tag{21}$$

or

$$\tilde{\mathcal{O}}\left(\frac{\Theta^2}{\rho^2}\left(\log\frac{1}{\Theta\nu} + \frac{\nu^2}{\varepsilon^2}\right)\left(\log\frac{|C|}{\rho\delta} + \log^2|C|\log^4\frac{1}{\Theta\nu}\right)\right) \tag{22}$$

### 4.3 Comparison to Replicability in Agnostic Passive Learning

We have an effective improvement of label complexity over the passive setting by having a multiplicative factor of $\Theta^2$ (for the first $N$ rounds) and $\frac{\nu^2}{\varepsilon^2}$ (for the last round) instead of the $\frac{1}{\varepsilon^2}$ factor in passive agnostic learning (equation 9). For distributions which are suitable for active learning (characterized by a low value of $\Theta$), and for problems with a reasonably low noise rate (characterized by a low value of $\nu$), both these values are much lower than $\frac{1}{\varepsilon^2}$.

## 5 Conclusions and Future Work

We presented the first *replicable* adaptations of two classical active-learning algorithms — CAL in the realizable setting and $A^2$ in the agnostic setting — yielding the RepliCAL and ReplicA$^2$ algorithms. By introducing randomized thresholding and replicable statistical-query subroutines, we show that one can retain the core label-complexity advantages of active learning under the strong stability requirement of replicability.

In the realizable case for finite hypothesis classes with suitable disagreement coefficients, RepliCAL matches the known dependence on $\Theta \log\frac{1}{\varepsilon}$ of CAL, incurring only a mild overhead for replicability. In the agnostic case, ReplicA$^2$ leverages the $A^2$ framework to handle noise and still improves over passive-learning bounds. These results demonstrate that, even under stringent stability constraints, adaptive querying can yield substantial label-complexity savings.

The transformation from replicability to differential privacy of Bun et al. (2023) continues to apply in the active-learning setting (though, notably, the reverse direction — from privacy to replicability —does not). This suggests that lower bounds for differentially private active learning may transfer to the replicable regime, offering a path to establishing tightness of our bounds. That said, we expect our sample complexity to be nearly tight, based on lower bounds in terms of $\rho$ and $|H|$ for replicable learning in the passive learning setting as well as lower bounds in terms of $\Theta$ and $\nu/\varepsilon$ for active learning without stability constraints.

A natural but challenging next step is to extend our results to infinite hypothesis classes. Standard active learning upper bounds in terms of VC dimension do not immediately carry over because private (hence replicable) learnability requires finite Littlestone dimension. It

would be valuable to show that finite Littlestone dimension, and therefore, global stability, still admits the active learning gains we obtain here.

Investigating a broader class of active learning algorithms, including those applicable to infinite hypothesis classes or structured prediction tasks, would be valuable future directions. Empirical studies will be essential to evaluate these methods in practical scenarios, providing further insights into their reliability and performance in real-world applications.

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

## A    Symbols

| Symbol | Description |
| --- | --- |
| $b$ | Random binary string |
| $c$ | Target function (realizable case) |
| $C$ | Hypothesis space |
| $d$ | Distance metric |
| $D$ | Probability distribution |
| $DIS(\cdot)$ | Disagreement region |
| $\delta$ | Failure probability of a model |
| $\Delta(\cdot)$ | Probability mass of disagreement region |
| $\varepsilon$ | Maximum error of returned hypothesis |
| err | Error of a hypothesis |
| $\text{err}_S$ | Empirical error of a hypothesis |
| $\text{err}_S^D$ | Empirical error of a hypothesis conditioned on disagreement region |
| $\phi$ | Query function in SQ learning |
| $h$ | Hypothesis function |
| $I$ | Threshold interval |
| $k$ | Number of samples |
| $m$ | Number of samples |
| $N$ | Maximum number of rounds the algorithm runs for |
| $\psi$ | Query function in SQ learning |
| $r$ | Round of algorithm |
| $R$ | Number of rounds of the algorithm for a specific run |
| $\rho$ | Replicability parameter |
| $S$ | Sample set |
| $T$ | Sample set of unlabeled points |
| $\Theta$ | Disagreement coefficient |
| $\nu$ | Noise |
| $v$ | Threshold for replicably discarding hypotheses |
| $\sigma$ | Threshold offset |
| $V$ | Version space |
| $\tau$ | Tolerance parameter in SQ learning |
| $\tau$ | Interval width in Replicable learning |
| $\pi$ | Global interval width in Replicable learning |
| $x$ | Sample |
| $y$ | Label |
| $\widehat{\cdot}$ | Empirical estimate |
| $\cdot'$ | Quantity of the last round |

## B    Replicable Active Realizable Learning

### B.1    Proof of Lemma 1

*Proof.* Proof of lemma 1 runs analogous to Balcan (2015). Letting $V_r$ denote the hypothesis space at round $r$, we will first argue convergence by showing that the distributional size of the disagreement region $\Delta(V_r)$ will be at least halved with each successive round, i.e., $\Delta(V_{r+1}) \leq \frac{\Delta(V_r)}{2}$ with high probability. Let $V_r^\Theta$ be the set of hypotheses in $V_r$ with large

error

$$V_r^\Theta = \left\{ h \in V_r : \text{err}(h) = d(h, c) \geq \frac{\Delta(V_r)}{2\Theta} \right\}. \tag{23}$$

If all hypotheses in this set are removed, the distributional size of the disagreement region will indeed be halved

$$\Delta(V_{r+1}) \leq \Delta\left( B\left( c, \frac{\Delta(V_r)}{2\Theta} \right) \right) \leq \Theta \frac{\Delta(V_r)}{2\Theta} = \frac{\Delta(V_r)}{2} \tag{24}$$

where the definition of the disagreement coefficient was used.

So as long as all high-error hypotheses are removed in each round, the size of the disagreement region is halved, and algorithm 3 converges in at most $\mathcal{O}\left( \log \frac{1}{\varepsilon} \right)$ steps, because the algorithm terminates when $\Delta(V_r) \leq \varepsilon$.

We now argue that with high probability, all high-error hypotheses are removed at each round. Note that because we are in the realizable setting, $\text{err}(h) = \Delta(V_r)\text{err}^{D_r}(h)$ for every $h \in V_r$, and so it follows that if $\Delta(V_r)\text{err}^{D_r}(h) \geq \frac{\Delta(V_r)}{2\Theta}$, then we can lower-bound the conditional error $\text{err}^{D_r}(h) \geq \frac{1}{2\Theta}$. It therefore suffices to remove all hypotheses with at least this conditional error on the disagreement region.

From algorithm 3 it is clear that since the hypotheses in each round are chosen to fall under a random threshold that is upper-bounded by $\frac{3}{8\Theta} - \frac{\tau}{2}$, this upper-bounds the conditional empirical error of the algorithm in each round. By applying Chernoff-Hoeffding bounds for the realizable case, we can bound the probability that any hypothesis with conditional error rate at least $\frac{1}{2\Theta}$ has empirical error rate less than $\frac{3}{8\Theta} - \frac{\tau}{2}$, for any of the $N = \mathcal{O}\left( \log \frac{1}{\varepsilon} \right)$ rounds of the algorithm. We see that the number of labeled points needed in each round to ensure good error estimates for all hypotheses with probability at least $1 - \frac{\delta}{2N}$ is :

$$\mathcal{O}\left( \Theta \log \frac{|C|N}{\delta} \right). \tag{25}$$

We take the sample for our empirical estimate of conditional error to be greater than this quantity, and so except with probability $\delta$, all high-error hypotheses are removed at every round. This guarantees convergence within $\mathcal{O}\left( \log \frac{1}{\varepsilon} \right)$ rounds, and so in total

$$\mathcal{O}\left( \Theta \log \frac{1}{\varepsilon} \log \frac{|C| \log \frac{1}{\varepsilon}}{\delta} \right) \tag{26}$$

labeled samples are required for convergence.

The size of the disagreement region can be estimated up to arbitrary accuracy using unlabeled data. Thus, we use the exact value in our algorithm.

It remains to argue the accuracy of the final hypothesis. A union bound over the failure probabilities of the empirical error rate estimation in each round yields an overall failure probability of $\mathcal{O}(\delta)$. Then, the stop condition $\Delta(V) \leq \varepsilon$ guarantees that all $h \in V_N$ have error rate below $\varepsilon$. This follows from the fact that in the realizable case, the ground truth $c$ will never be removed from the hypothesis space because the estimated error rate of the ground truth cannot exceed 0. Since the ground truth $c$ is never removed, if all hypotheses agree on a point, all of them must classify this point correctly. The final thresholding after exiting the loop is added for the purpose of replicability and does not have an adverse effect on accuracy.

$\square$

## B.2 Proof of Lemma 2

*Proof.* Let the RepliCAL algorithm be run on two different ordered sets of samples $S^1 = \bigcup_{r=1}^{R+1} S_r^1$ and $S^2 = \bigcup_{r=1}^{R+1} S_r^2$ drawn from the respective distributions $\{D_1^1, \ldots, D_{R+1}^1\}$ and $\{D_1^2, \ldots, D_{R+1}^2\}$, which are obtained by conditioning the distribution $D$ on the disagreement region of the corresponding round $(1, \ldots, R+1)$.

Select an interval width $\pi \leq \mathcal{O}\left(\frac{\varepsilon \rho^2}{\Theta \log |C|}\right)$ which divides $\frac{\varepsilon}{8\Theta}$. Define $I_i$ to be intervals corresponding to the desired global error rate in the final thresholding round

$$
\begin{aligned}
I_0 &= [0, \pi) \\
I_1 &= [\pi, 2\pi) \\
&\;\;\vdots \\
I_{\frac{\varepsilon}{8\Theta\pi}} &= \left[\frac{\varepsilon}{8\Theta} - \pi, \frac{\varepsilon}{8\Theta}\right)
\end{aligned}
\tag{27}
$$

and $v_i' \Delta(V_{R+1}) = \frac{2i+1}{2} \cdot \pi$ to be the respective global thresholds.

Let $V^1(i)$ and $V^2(i)$ denote the two final version spaces across the two independent sets of samples $S^1$ and $S^2$ and for a shared randomly chosen threshold $v_i'$. In the following proof we will drop the explicit dependence on $(i)$ for conciseness.

We will show that with probability at least $1 - \frac{\rho}{8}$, for $S^1$ and $S^2$ each of size $\tilde{\mathcal{O}}\left(\Theta \log \frac{1}{\varepsilon} \cdot \frac{\log^2 |C| \log \frac{1}{\rho} \log^4 \frac{1}{\varepsilon}}{\rho^4}\right)$ we have:

$$
\frac{|V^1 \Delta V^2|}{|V^1 \cup V^2|} \leq \frac{\rho}{4}.
\tag{28}
$$

To prove the claim, we, analogous to Bun et al. (2023), call a threshold $v_i'$ "bad" if any of the following conditions hold:

1. The $i$th interval has too many elements:

$$
|I_i| > \frac{\rho}{30} \left|I_{[i-1]}\right|.
\tag{29}
$$

2. The number of elements beyond $I_i$ increases too quickly:

$$
\exists j \geq 1 : |I_{i+j}| \geq e^j \left|I_{[i-1]}\right|.
\tag{30}
$$

and "good" otherwise.

Here, $|I_i|$ denotes the number of hypotheses whose true risk lies in interval $I_i$, and $\left|I_{[i]}\right|$ the number of hypotheses in intervals up through $I_i$.

We will be proving the following:

1. If $v_i'$ is a good threshold, then $V^1$ and $V^2$ are probably close

$$
\Pr_{S_1, S_2}\left[\frac{|V^1 \Delta V^2|}{|V^1 \cup V^2|} \leq \frac{\rho}{4}\right] \geq 1 - \frac{\rho}{8}.
\tag{31}
$$

2. At most a $\frac{\rho}{8}$ fraction of thresholds are bad.

**Part 1**  To prove the first part, we consider three cases in which mistakes can occur. For this, we define a "good" hypothesis as a hypothesis with a global true error rate less than $\Delta(V_{R+1})v_i'$ and call it "bad" otherwise

1. A "bad" hypothesis with $\mathrm{err}(h) \in I_{i+j}$ was accepted in every round.

2. A "good" hypothesis with $\mathrm{err}(h) \in I_{i-j}$ was rejected in any round.

3. For any hypothesis in the last round with $\mathrm{err}(h) \in I_i$, the empirical error is on the wrong side of the threshold $\Delta(V_{R+1})v_i'$.

By a Chernoff bound, the probability of a hypothesis with true global error rate $\mathrm{err}(h) \in I_{i+j}, j > 0$ having an empirical error rate less than $\Delta(V_{R+1})v_i'$ after the final thresholding is

$$\Pr\left[\mathrm{err}_{S_{R+1}}^{D_{R+1}}(h) \leq v_i'\right] \leq \mathrm{e}^{-\Omega\left(\frac{(j\tau')^2|S_{R+1}|}{(i+j)\tau'}\right)} \leq \mathrm{e}^{-\Omega(j^2\tau'^2\Theta k_N)} \tag{32}$$

where $k_N = |S_r|$ and the conditional error rate is computed by scaling the global error rate by the size of the disagreement region. The estimation tolerance is of the order of the global interval width scaled by the disagreement region

$$\frac{\tau'}{2} = \frac{\pi}{2\Delta(V_{R+1})} = \mathcal{O}\left(\frac{\varepsilon\rho^2}{\Delta(V_{R+1})\Theta\log|C|}\right) \geq \mathcal{O}\left(\frac{\rho^2}{\Theta\log|C|}\right). \tag{33}$$

The probability of the first case occurring is upper-bounded by this Chernoff bound for any single round. For simplicity, here we choose the last round $r = R + 1$. We introduce the random variable $x_i$ that counts the number of hypotheses with $\mathrm{err}(h) \in I_{i+j}, j > 0$ which cross the threshold $v_i'$ in the final round. Then, assuming the chosen threshold is good, the expected value can be bounded by

$$\begin{aligned}\mathbb{E}[x_i] &\leq \sum_{j>0} |I_{i+j}|\, \mathrm{e}^{-\Omega(j^2\tau'^2\Theta k_N)} \\ &\leq |I_{[i-1]}| \sum_{j>0} \mathrm{e}^{-\Omega(j^2\log 1/\rho - j)} \leq |I_{[i-1]}| \sum_{j>0} \rho^{\mathcal{O}(j^2)} \\ &\leq \frac{\rho^2}{30\cdot 64} |I_{[i-1]}|.\end{aligned} \tag{34}$$

Here, the second condition for good thresholds and size of the samples $k_N$ was used. The last step follows from an asymptotic consideration that holds for small enough constants. Using Markov's inequality, we conclude that

$$\Pr\left[x_i \geq \frac{\rho}{30}|I_{[i-1]}|\right] \leq \frac{\rho}{64}. \tag{35}$$

For the second case, the probability of one good hypothesis — measured by the last round — crossing the threshold in any round is given by a union bound over all rounds. The conditional error rates of any good hypothesis in rounds $r \leq R + 1$ where $\Delta(V_r) \geq \varepsilon$ will be lower or equal than the threshold $v_i$:

$$\mathrm{err}(h) \leq \Delta(V_{R+1})v_i' \tag{36}$$
$$\mathrm{err}^{D_r}(h) \leq \frac{\Delta(V_{R+1})}{\Delta(V_r)}v_i' = \frac{\Delta(V_{R+1})}{\Delta(V_r)} \cdot \frac{\varepsilon}{\Delta(V_{R+1})}v_i \tag{37}$$
$$\leq v_i \tag{38}$$

Therefore, the probability of the error rate estimate crossing the threshold can be upper-bounded by the final round $R + 1$

$$\Pr\left[\mathrm{err}_{S_{R+1}}^{D_{R+1}}(h) \leq v_i'\right] \leq N\mathrm{e}^{-\Omega(j^2\tau'^2\Theta k_N)}. \tag{39}$$

Defining random variable $y_i$ to be the number of good hypotheses rejected at any round, we have

$$\mathbb{E}[y_i] \leq N \sum_{j>0} |I_{i-j}| \, e^{-\Omega(j^2 \tau'^2 \Theta k_N)} \tag{40}$$

$$\leq |I_{[i-1]}| \, N e^{-\Omega(\tau'^2 \Theta k_N)} \leq |I_{[i-1]}| \, N \rho^{\mathcal{O}(j^2)}.$$

Again, we conclude that with probability at least $1 - \frac{\rho}{64}$ only a $\frac{\rho}{30}$ fraction of hypotheses will fall under case 2.

In the third case, by definition of the first condition for bad thresholds, we directly see that in the worst case the number of hypotheses is upper-bounded by the number of hypotheses in the interval

$$|I_i| \leq \frac{\rho}{30} |I_{[i-1]}|. \tag{41}$$

Thus, in total there will be no more than $\frac{\rho}{10} |I_{[i-1]}|$ mistakes made with high probability $1 - \frac{\rho}{32}$. Considering two different runs of the algorithm, the symmetric difference of the final hypothesis sets will be less than $\frac{\rho}{5} |I_{[i-1]}|$ with high probability at least $1 - \frac{\rho}{16}$.

Furthermore, the union of the sets is guaranteed to be at least $\left(1 - \frac{\rho}{15}\right) |I_{[i-1]}|$ with a failure probability of at most $1 - \frac{\rho}{32}$ as seen in the analysis of the second case.

Finally, a union bound yields the desired result

$$\Pr_{S^1, S^2} \left[ \frac{|V^1 \Delta V^2|}{|V^1 \cup V^2|} \leq \frac{\rho}{4} \right] \geq 1 - \frac{\rho}{8}. \tag{42}$$

**Part 2** Now, let us prove that almost all thresholds are good. The structure of the proof is based on lower-bounding the number of hypotheses "contributed" by each of the "bad" intervals, which in turn upper-bounds the number of "bad" intervals, since the total number of hypotheses is fixed by the size of the concept class, $|C|$.

Let the "bad" intervals be present in $\ell$ clusters (longest consecutive "bad" intervals bounded by "good" interval(s)), with the $j^{th}$ "bad" cluster containing $t_j$ continuous "bad" intervals. Thus, the total number of "bad" intervals is $\sum_{j=1}^{\ell} t_j$.

First, let's say the intervals are "bad" by condition 1 of "badness". Then the $j^{th}$ bad cluster increases the number of hypotheses from $I_{[i_j]}$ by at least $\left(1 + \frac{\rho}{30}\right)^{t_j}$.

If the intervals are bad by condition 2 of "badness", the $j^{th}$ cluster increases the number of hypotheses (corresponding to the future interval(s) causing them to be "bad" by condition 2), by at least $e^{t_j}$. Since $e > \left(1 + \frac{\rho}{30}\right)$, the statement that the $j^{th}$ cluster increases the number of hypotheses by "at least" $\left(1 + \frac{\rho}{30}\right)^{t_j}$ still holds, for both conditions of "badness". Since $|I_0| > 1$, we can write:

$$|C| \geq \left(1 + \frac{\rho}{30}\right)^{\sum_{j=1}^{\ell} t_j} \tag{43}$$

It follows that the number of "bad" intervals:

$$\sum_{j=1}^{\ell} t_j \leq \mathcal{O}\left(\frac{\log |C|}{\rho}\right) \tag{44}$$

Since $\tau'$ has been chosen such that the total number of intervals is at least $\mathcal{O}\left(\frac{\log |C|}{\rho^2}\right)$, the fraction of intervals that are "bad" is $\mathcal{O}(\rho)$

This is true for each round of our algorithm. If we want to bound the probability of choosing a bad interval in "any" round, we have to take a union bound of the probability of bad intervals in each round. By choosing $\rho' = \frac{\rho}{N}$ where $\rho$ is the replicability-factor of the parent

algorithm, and using an appropriate constant, we can union-bound over $N$ rounds to have the probability over all rounds to be $\frac{\rho}{8}$. This requirement of having to choose a smaller $\rho$ will be accounted for while calculating the label complexity.

Three events can break replicability of the proposed algorithm: A bad interval is randomly selected, the sets $V^1$ and $V^2$ are not close or two different random hypotheses are chosen even though the final sets are close. The probabilities of these bad events occurring are $\frac{\rho}{8}$, $\frac{\rho}{8}$, and $\frac{\rho}{4}$ respectively. Thus, a union bound yields a failure probability of at most $\frac{\rho}{2}$, satisfying $\rho$-replicability as required. $\qquad\square$

### B.3  Proof of Theorem 1

*Proof.* From the proof of lemma 1, we have bounds on the number of samples required for algorithm 3 to get an error rate of at most $\varepsilon$ with high probability $1 - \delta$, and converge within $\mathcal{O}\left(\log \frac{1}{\varepsilon}\right)$ rounds.

Furthermore, in lemma 2, we have seen the worst case sample complexity for the thresholding to be $\rho$-replicable is $k_N = \mathcal{O}\left(\frac{\log \frac{1}{\rho}}{\Theta \tau'^2}\right)$. Since $\tau' \leq \mathcal{O}\left(\frac{\rho^2}{\Theta \log |C|}\right)$, we can replace $\tau'$ to get sample size as:

$$k_N = \mathcal{O}\left(\frac{\Theta \log^2 |C| \log \frac{1}{\rho}}{\rho^4}\right). \tag{45}$$

This is the label complexity required in each round. Hence, the total label complexity required for $\rho$-replicability after $N$ rounds is

$$\mathcal{O}\left(N \cdot \frac{\Theta \log^2 |C| \log \frac{1}{\rho}}{\rho^4}\right). \tag{46}$$

While proving lemma 2, we stated that in order for the algorithm to be $\rho$-replicable, the thresholding subroutine has to be run with a lower replicability parameter: $\frac{\rho}{N}$, where $N$ is the number of rounds. Hence, the corresponding label complexity should be corrected to:

$$\mathcal{O}\left(N \cdot \frac{\Theta \log^2 |C| \log \frac{N}{\rho} N^4}{\rho^4}\right). \tag{47}$$

Lemma 1 states that the number of rounds required for convergence is $\mathcal{O}\left(\log \frac{1}{\varepsilon}\right)$. Hence, the label complexity is

$$\mathcal{O}\left(\Theta \log \frac{1}{\varepsilon} \cdot \frac{\log^2 |C| \log \frac{\log \frac{1}{\varepsilon}}{\rho} \log^4 \frac{1}{\varepsilon}}{\rho^4}\right). \tag{48}$$

The label complexity required to ensure bounded error as well as replicability can be found by combining equation 26 and equation 48. The overall complexity thus derived is:

$$\mathcal{O}\left(\Theta \log \frac{1}{\varepsilon} \cdot \frac{\log^2 |C| \log \frac{\log \frac{1}{\varepsilon}}{\rho} \log^4 \frac{1}{\varepsilon}}{\rho^4} + \Theta \log \frac{1}{\varepsilon} \left[\log \frac{|C| \log \frac{1}{\varepsilon}}{\delta}\right]\right). \tag{49}$$

This gives us the required label complexity

$$\mathcal{O}\left(\Theta \log \frac{1}{\varepsilon} \cdot \frac{\log^2 |C| \log \frac{\log \frac{1}{\varepsilon}}{\rho} \log^4 \frac{1}{\varepsilon} + \rho^4 \log \frac{|C| \log \frac{1}{\varepsilon}}{\delta}}{\rho^4}\right), \tag{50}$$

as stated in theorem 1, and concludes the proof.

It can be argued that $\log \log \frac{1}{\varepsilon}$ is trivial w. r. t. the other terms, and the label complexity thus reduces to

$$\tilde{\mathcal{O}}\left(\Theta \log \frac{1}{\varepsilon} \cdot \frac{\log^2 |C| \log^4 \frac{1}{\varepsilon} + \rho^4 \log \frac{|C|}{\delta}}{\rho^4}\right). \tag{51}$$

$\square$

## C  Replicable Active Agnostic Learning

### C.1  Proof of Lemma 3

*Proof.* Our proof runs analogous to Balcan (2015). Let $V_r$ denote the hypothesis space at round $r$. The distributional size of the disagreement region $\Delta(V_r)$ will be at least halved with each successive round $\Delta(V_{r+1}) \le \Delta(V_r)/2$ with high probability. Let $V_r^{\Theta}$ be the set of hypotheses in $V_r$ with large error

$$V_r^{\Theta} = \left\{h \in V_r : d(h, h^*) \ge \frac{\Delta(V_r)}{2\Theta}\right\}. \tag{52}$$

If all hypotheses in this set are removed, the distributional size of the disagreement region will indeed be halved

$$\Delta(V_{r+1}) \le \Delta\left(B\left(h^*, \frac{\Delta(V_r)}{2\Theta}\right)\right) \le \Theta \frac{\Delta(V_r)}{2\Theta} = \frac{\Delta(V_r)}{2} \tag{53}$$

where the definition of the disagreement coefficient was used.

Since the size of the disagreement region is halved in each round with high probability, and the loop stops when $\Delta(V_r) \le 8\Theta\nu$, the convergence would take at most $N \in \mathcal{O}\left(\log \frac{1}{\Theta\nu}\right)$ steps.

First, we show that in the round based portion of the algorithm, hypotheses in the set $V_r^{\Theta}$ will be removed with high probability. From the definition of the distance metric we get

$$\begin{aligned} d(h, h^*) &= \Delta(V_r) \Pr_{x \sim D_r}[h(x) \ne h^*(x)] \\ &\le \Delta(V_r)\left[\mathrm{err}^{D_r}(h) + \mathrm{err}^{D_r}(h^*)\right] \\ &\le \Delta(V_r)\mathrm{err}^{D_r}(h) + \nu. \end{aligned} \tag{54}$$

Assuming that $h \in V_r^{\Theta}$ we get

$$\begin{aligned} &\Delta(V_r)\mathrm{err}^{D_r}(h) \ge d(h, h^*) - \nu \\ &\Rightarrow \mathrm{err}^{D_r}(h) \ge \frac{1}{2\Theta} - \frac{\nu}{\Delta(V_r)} \\ &\qquad\qquad \ge \frac{1}{2\Theta} - \frac{1}{8\Theta} = \frac{3}{8\Theta} \end{aligned} \tag{55}$$

using $\frac{1}{\Delta(V_r)} \le \frac{1}{8\Theta\nu}$. This in turn implies that the empirical conditional error $\mathrm{err}_{S_r}^{D_r}(h)$, which is estimated to within tolerance $\frac{1}{16\Theta}$, must be greater than $\frac{5}{16\Theta}$.

Recall that the largest value the empirical error threshold for removing a hypothesis, $v + \sigma_r$, can take is

$$\frac{\nu}{\Delta(V_r)} + \frac{3}{16\Theta} - \frac{\tau}{2} < \frac{\nu}{8\Theta\nu} + \frac{3}{16\Theta} = \frac{5}{16\Theta}. \tag{56}$$

Therefore

$$\mathrm{err}_{S_r}^{D_r}(h) \ge \mathrm{err}^{D_r}(h) - \frac{1}{16\Theta} \ge \frac{5}{16\Theta} \ge v + \sigma_r \tag{57}$$

and therefore we will remove $h$ with probability at least $1 - \frac{\delta}{2(N+1)}$. This argument relies on estimating the conditional error of every hypothesis in $C$ to within tolerance $\frac{1}{16\Theta}$, and so we require labels for

$$\mathcal{O}\left(\Theta^2 \log \frac{|C|N}{\delta}\right) \tag{58}$$

points sampled i.i.d. from $\Delta(V_r)$, by a Chernoff bound. Thus, the total sample complexity for all $N$ rounds is

$$k = \mathcal{O}\left(N\Theta^2 \log \frac{|C|N}{\delta}\right) \tag{59}$$

To see that with high probability the best hypothesis $h^*$ is never removed from the version space $V$, observe that the smallest value the empirical error threshold for removing a hypothesis, $v + \sigma_r$, can take is

$$\frac{\nu}{\Delta(V_r)} + \frac{1}{16\Theta} + \frac{3\tau}{2} > \frac{\nu}{\Delta(V_r)} + \frac{1}{16\Theta}. \tag{60}$$

The error of every hypothesis is estimated to within $\frac{1}{16\Theta}$ with high probability, so we have that

$$\mathrm{err}_{S_r}^{D_r}(h^*) \leq \frac{\nu}{\Delta(V_r)} + \frac{1}{16\Theta}, \tag{61}$$

and therefore $h^*$ is never removed with high probability.

Now we consider the accuracy of the hypothesis returned at the end of the algorithm. We may assume, because the loop has terminated, that $\Delta(V_{R+1}) \leq 8\Theta\nu$.

$$\begin{aligned}
\mathrm{err}(h) - \mathrm{err}(h^*) &= \Delta(V_{R+1})\left[\mathrm{err}^{D_{R+1}}(h) - \mathrm{err}^{D_{R+1}}(h^*)\right] \\
&\leq 8\Theta\nu\left[\mathrm{err}^{D_{R+1}}(h) - \mathrm{err}^{D_{R+1}}(h^*)\right]
\end{aligned} \tag{62}$$

Therefore, it suffices to find a hypothesis with $\mathrm{err}^{D_{R+1}}(h) \leq \mathrm{err}^{D_{R+1}}(h^*) + \frac{\varepsilon}{8\Theta\nu}$ to ensure $\mathrm{err}(h) \leq \mathrm{err}(h^*) + \varepsilon$. We estimate the conditional error of every hypothesis on the last disagreement region with an accuracy of $\frac{\varepsilon}{192\Theta\nu}$ and failure probability of $\frac{\delta}{2(N+1)}$. This requires a sample set size of

$$\mathcal{O}\left(\Theta^2 \frac{\nu^2}{\varepsilon^2} \log \frac{N|C|}{\delta}\right). \tag{63}$$

Defining $\hat{\nu}^{D_{R+1}}$ as the minimum of all estimated error rates ensure that the optimal conditional error rate is estimated within a tolerance of $\frac{\varepsilon}{192\Theta\nu}$.

The largest threshold for the final round is

$$\hat{\nu}^{D_{R+1}} + \frac{2\varepsilon}{96\Theta\nu} - \frac{\tau'}{2} \leq \mathrm{err}^{D_{R+1}}(h^*) + \frac{\varepsilon}{192\Theta\nu} + \frac{2\varepsilon}{96\Theta\nu} \tag{64}$$

Therefore any bad hypothesis $h$ will be removed

$$\begin{aligned}
\mathrm{err}_{S_r}^{D_{R+1}}(h) &\geq \mathrm{err}^{D_{R+1}}(h) - \frac{\varepsilon}{192\Theta\nu} \\
&\geq \mathrm{err}^{D_{R+1}}(h^*) + \frac{\varepsilon}{8\Theta\nu} - \frac{\varepsilon}{192\Theta\nu} \\
&\geq v + \hat{\nu}^{D_{R+1}}.
\end{aligned} \tag{65}$$

Proving that the optimal hypothesis is never removed also proves that the version space will never be empty after the final round. This is guaranteed because

$$\begin{aligned}
\mathrm{err}_{S_{R+1}}^{D_{R+1}}(h^*) &\leq \mathrm{err}^{D_{R+1}}(h^*) + \frac{\varepsilon}{192\Theta\nu} \\
&\leq \hat{\nu}^{D_{R+1}} + \frac{\varepsilon}{192\Theta\nu} + \frac{\varepsilon}{192\Theta\nu}
\end{aligned} \tag{66}$$

is less or equal than the smallest threshold.

So overall, combining equation 59 and equation 63, and substituting $N$ we get that

$$\mathcal{O}\left(\Theta^2 \left(\log \frac{1}{\Theta\nu} \log \frac{|C| \log \frac{1}{\delta}}{\delta} + \frac{\nu^2}{\varepsilon^2} \log \frac{\log \frac{1}{\Theta\nu}|C|}{\delta}\right)\right) \tag{67}$$

many labeled samples are required for convergence to a good hypothesis.

$\square$

## C.2  Proof of Lemma 4

*Proof.* Let the RepllcA$^2$ algorithm be run on two different ordered sets of samples $S^1 = \bigcup_{r=1}^{R+1} S_r^1$ and $S^2 = \bigcup_{r=1}^{R+1} S_r^2$ drawn from the respective distributions $\{D_1^1, \dots, D_{R+1}^1\}$ and $\{D_1^2, \dots, D_{R+1}^2\}$, which are obtained by conditioning the distribution $D$ on the disagreement region $V_i$ of the corresponding round $(1, \dots, R+1)$.

Select an interval width $\pi \leq \mathcal{O}\left(\frac{\varepsilon\rho^2}{\log |C|}\right)$, which should divide $\frac{\varepsilon}{12}$. Define $I_i$ to be intervals corresponding to the global error rate in the last round

$$I_0 = \left[\frac{\varepsilon}{12}, \frac{\varepsilon}{12} + \pi\right)$$

$$I_1 = \left[\frac{\varepsilon}{12} + \pi, \frac{\varepsilon}{12} + 2\pi\right) \tag{68}$$

$$\vdots$$

$$I_{\frac{\varepsilon}{12\pi}} = \left[\frac{2\varepsilon}{12} - \pi, \frac{2\varepsilon}{12}\right)$$

and $v_i' \Delta(V_{R+1}) = \frac{\varepsilon}{12} + \frac{2i+1}{2} \cdot \pi$ be the respective thresholds.

Let $V^1(i)$ and $V^2(i)$ denote the two final version spaces across the two independent sets of samples $S^1$ and $S^2$ and for a shared randomly chosen threshold $v_i'$.

We prove that with probability at least $1 - \frac{\rho}{8}$, for samples $S^1$ and $S^2$ drawn i.i.d from $D_r$, each of size $\tilde{\mathcal{O}}\left(\frac{\Theta^2 \log^2 |C| \log \frac{1}{\rho} \log^4 \frac{1}{\Theta\nu}}{\rho^4} \left(\log \frac{1}{\Theta\nu} + \frac{\nu^2}{\varepsilon^2}\right)\right)$, we have:

$$\frac{|V^1 \Delta V^2|}{|V^1 \cup V^2|} \leq \frac{\rho}{4}. \tag{69}$$

Analogously to the realizable case, we define "good" and "bad" thresholds. As before, we will be proving the following:

1. If $v_i'$ is a good threshold, then $V_1$ and $V_2$ are probably close

$$\Pr_{S_1, S_2}\left[\frac{|V_1 \Delta V_2|}{|V_1 \cup V_2|} \leq \frac{\rho}{4}\right] \geq 1 - \frac{\rho}{8}. \tag{70}$$

2. At most a $\frac{\rho}{8}$ fraction of thresholds are bad.

**Part 1**  To prove the first part, we consider three cases in which mistakes can occur. For the following analysis we will define a hypothesis as "good" if it has a global true error rate less than $\nu + \Delta(V_{R+1})v_i'$ and define it as "bad" otherwise.

1. A "bad" hypothesis with $\text{err}(h) - \nu \in I_{i+j}$ was accepted in every round, i.e., with empirical error smaller than the threshold $v_i$.

2. A "good" hypothesis in the last round with $\text{err}(h) - \nu \in I_{i-j}$ was rejected in any round, i.e., with empirical error larger than the threshold $v_i$.

3. For any hypothesis in the last round with $\text{err}(h) - \nu \in I_i$, the empirical error is on the wrong side of the threshold $\nu + \Delta(V_{R+1})v_i'$.

By a Chernoff bound, the probability of a hypothesis with true global error rate $\text{err}(h) - \nu \in I_{i+j}, j > 0$ having an empirical error rate less than $\widehat{\nu}^D + v_i'$ after the final thresholding is at most

$$\Pr\left[\text{err}_{S_{R+1}}^{D_{R+1}}(h) \leq \widehat{\nu}^{D_{R+1}} + v_i'\right] \leq e^{-\Omega\left(j^2\tau'^2|S_{R+1}|\right)} = e^{-\Omega(j^2\tau'^2 k_N)} \tag{71}$$

where $k_N = |S_r|$. The estimation tolerance is of the order of the global interval width scaled by the disagreement region

$$\frac{\tau'}{4} = \frac{\pi}{4\Delta(V_{R+1})} = \mathcal{O}\left(\frac{\varepsilon\rho^2}{\Delta(V_{R+1})\log|C|}\right) \geq \mathcal{O}\left(\frac{\varepsilon\rho^2}{\Theta\nu\log|C|}\right) \tag{72}$$

This ensures that a hypothesis with estimation error $\frac{\tau'}{4}$ does not cross the threshold which itself depends on the optimal hypothesis estimated to within an error of $\frac{\tau'}{4}$. The probability of the first case occurring is upper-bounded by this Chernoff bound for any single round. For simplicity, here we chose the final thresholding $r = R + 1$. We introduce the random variable $x_i$ that counts the number of hypotheses with $\text{err}(h) - \nu \in I_{i+j}, j > 0$ which cross the threshold $v_i'$ in the last round. Then, the expected value can be bounded by — assuming that the chosen threshold is good

$$\begin{aligned}
\mathbb{E}[x_i] &\leq \sum_{j>0} |I_{i+j}| e^{-\Omega(j^2\tau'^2 k_N)} \\
&\leq |I_{[i-1]}| \sum_{j>0} e^{-\Omega(j^2\log 1/\rho - j)} \leq |I_{[i-1]}| \sum_{j>0} \rho^{\mathcal{O}(j^2)} \\
&\leq \frac{\rho^2}{30 \cdot 64} |I_{[i-1]}|.
\end{aligned} \tag{73}$$

Here, the second condition for good thresholds and size of the samples $k$ was used. The last step follows from an asymptotic consideration that holds for small enough constants. Using Markov's theorem, we conclude that

$$\Pr\left[x_i \geq \frac{\rho}{30}|I_{[i-1]}|\right] \leq \frac{\rho}{64}. \tag{74}$$

For the second case, the probability of one good hypothesis — measured by the last round — crossing the threshold in any round is given by a union bound over all rounds. The conditional error rates of any good hypothesis in rounds $r < R + 1$ where $\Delta(V_r) > 8\Theta\nu$ will be lower or equal than the respective thresholds $\sigma_r + v_i$ under the benign assumption that $\varepsilon < \nu$

$$\text{err}(h) \leq \nu + \Delta(V_{R+1})v_i' \tag{75}$$

$$\text{err}^{D_r}(h) \leq \nu^{D_r} + \frac{\Delta(V_{R+1})}{\Delta(V_r)}v_i' \tag{76}$$

$$\leq \nu^{D_r} + \frac{\Delta(V_{R+1})}{\Delta(V_r)} \cdot \frac{\varepsilon}{\nu}v_i \tag{77}$$

$$\leq \frac{\nu}{\Delta(V_r)} + v_i. \tag{78}$$

Therefore, the probability of the error rate estimate crossing the threshold can be upper-bounded by the final round $R + 1$

$$\Pr\left[\text{err}_{S_{N+1}}^{D_{N+1}}(h) \leq \widehat{\nu}^{D_{R+1}} + v_i'\right] \leq N e^{-\Omega(j^2\tau'^2 k_N)} \tag{79}$$

and the expectation of the random variable analogous to $x_i$ defined as $y_i$ is upper-bounded by

$$\mathbb{E}[y_i] \leq N \sum_{j>0} |I_{i-j}| \, \mathrm{e}^{-\Omega(j^2 \tau'^2 k_N)}$$

$$\leq \left|I_{[i-1]}\right| N \mathrm{e}^{-\Omega(\tau'^2 \Theta k_N)} \leq \left|I_{[i-1]}\right| N \rho^{\mathcal{O}(j^2)}. \tag{80}$$

Again, we conclude that with probability at least $1 - \frac{\rho}{64}$ only a fraction of $\frac{\rho}{30}$ hypotheses will have made a mistake according to case 2.

As before, in the third case the number of hypotheses is upper-bounded by the number of hypotheses in the interval by the definition of the first condition of bad thresholds

$$|I_i| \leq \frac{\rho}{30} |I_{[i-1]}|. \tag{81}$$

Thus, in total, there will be no more than $\frac{\rho}{10}|I_{[i-1]}|$ mistakes made with high probability $1 - \frac{\rho}{32}$. Considering two different runs of the algorithm, the symmetric difference of the final hypothesis sets will be less than $\frac{\rho}{5}|I_{[i-1]}|$ with high probability at least $1 - \frac{\rho}{16}$.

Furthermore, the union of the sets is guaranteed to be at least $\left(1 - \frac{\rho}{15}\right)|I_{[i-1]}|$ with a failure probability of at most $1 - \frac{\rho}{32}$ as seen in the analysis of the second case.

Finally, a union bound yields the desired result

$$\Pr_{S^1, S^2} \left[ \frac{\left|V_1^{(i)} \Delta V_2^{(i)}\right|}{\left|V_1^{(i)} \cup V_2^{(i)}\right|} \leq \frac{\rho}{4} \right] \geq 1 - \frac{\rho}{8}. \tag{82}$$

**Part 2** Proving that almost all thresholds are "good" follows the same argument as in the realizable setting, and we can conclude that the fraction of intervals that are "bad" is $\mathcal{O}(\rho)$.

This is true for each round $n = 1, 2, ..., N+1$ of our algorithm. By choosing $\rho' = \frac{\rho}{2(N+1)}$ where $\rho$ is the replicability-factor of the parent algorithm, and using an appropriate constant, we can union-bound over $N$ rounds to have the probability over all rounds to be $\frac{\rho}{8}$. Union-bounding over the "bad" events gives us a total failure probability of $\frac{\rho}{2}$, as in the realizable setting, hence proving $\rho$-replicability as required. $\qquad\square$

### C.3   Proof of Theorem 2

*Proof.* Equation 26 gives us the worst-case sample complexity of our algorithm required to get an error rate of at most $\nu + \varepsilon$ with high probability $1 - \delta$.

Furthermore, in the proof of lemma 4, we have seen the worst case sample complexity for the thresholding to be $\rho$-replicable is $k_N = \mathcal{O}\left(\frac{\log \frac{1}{\rho}}{\tau^2}\right)$. Since $\tau \leq \mathcal{O}\left(\frac{\rho^2}{\Theta \log |C|}\right)$, we can replace $\tau$ to get sample size as:

$$k_N = \mathcal{O}\left(\frac{\Theta^2 \log^2 |C| \log \frac{1}{\rho}}{\rho^4}\right). \tag{83}$$

This is the label complexity required in each round. Hence, the total label complexity required for $\rho$-replicability after $N$ rounds is

$$\mathcal{O}\left(N \cdot \frac{\Theta^2 \log^2 |C| \log \frac{1}{\rho}}{\rho^4}\right). \tag{84}$$

While proving lemma 4, we stated that in order for the algorithm to be $\rho$-replicable, the thresholding subroutine has to be run with a lower replicability parameter of the order of $\frac{\rho}{N}$. Hence, the corresponding label complexity should be corrected to:

$$\mathcal{O}\left(N \cdot \frac{\Theta^2 \log^2 |C| \log \frac{N}{\rho} N^4}{\rho^4}\right). \tag{85}$$

Lemma 1 states that the number of rounds required for convergence is $\mathcal{O}\left(\log \frac{1}{\Theta\nu}\right)$. Hence, the label complexity is

$$\mathcal{O}\left(\Theta^2 \log \frac{1}{\Theta\nu} \cdot \frac{\log^2 |C| \log \frac{\log \frac{1}{\Theta\nu}}{\rho} \log^4 \frac{1}{\Theta\nu}}{\rho^4}\right). \tag{86}$$

To ensure $\rho$-replicability in the last round we need $\mathcal{O}\left(\frac{\log \frac{1}{\rho}}{\tau'^2}\right)$ labels. To ensure the same number of intervals, the replicability-constant should be the same as the one before, $\frac{\rho}{N}$. Since $\tau' \leq \mathcal{O}\left(\frac{\varepsilon\rho^2}{\Theta\nu \log |C|}\right)$, we have the label complexity in the last round as

$$\mathcal{O}\left(\Theta^2 \frac{\nu^2}{\varepsilon^2} \cdot \frac{\log^2 |C| \log \frac{\log \frac{1}{\Theta\nu}}{\rho} \log^4 \frac{1}{\Theta\nu}}{\rho^4}\right) \tag{87}$$

The label complexity required to ensure bounded error as well as replicability can be found by combining equation 67, equation 86 and equation 87. The overall complexity thus derived is:

$$\mathcal{O}\Bigg(\Theta^2\Bigg(\left(\log \frac{1}{\Theta\nu} + \frac{\nu^2}{\varepsilon^2}\right)\frac{\log^2 |C| \log \frac{\log \frac{1}{\Theta\nu}}{\rho} \log^4 \frac{1}{\Theta\nu}}{\rho^4}$$
$$+ \log \frac{1}{\Theta\nu} \log \frac{|C| \log \frac{1}{\Theta\nu}}{\delta} + \frac{\nu^2}{\varepsilon^2} \log \frac{|C|}{\delta}\Bigg)\Bigg) \tag{88}$$

or

$$\tilde{\mathcal{O}}\left(\Theta^2 \left(\log \frac{1}{\Theta\nu} + \frac{\nu^2}{\varepsilon^2}\right)\left(\log \frac{|C|}{\delta} + \frac{\log^2 |C| \log^4 \frac{1}{\Theta\nu}}{\rho^4}\right)\right) \tag{89}$$

$\square$

