# OpenReview forum: "The Cost of Replicability in Active Learning"
_TMLR — Accepted by TMLR_

### Review · Reviewer_bvPC · 2025-09-17

**Summary Of Contributions:**

Summary:


This paper studies replicable learning in the context of active learning. It presents the first replicable learning algorithms for finite hypothesis classes in both the realizable and agnostic settings. The proposed methods are augmentations of CAL and A², enhanced with the randomized threshold trick of Bun et al. (2023). The authors provide label complexity bounds for these algorithms and compare them to replicable learners in the passive setting. In certain regimes—depending on the disagreement coefficient—the active learner achieves substantial label savings.



Strengths:
1. The paper addresses an interesting problem. Active learning is a classical topic in learning theory, while replicable learning has gained significant attention in recent years. Combining the two opens the door to potentially impactful applications.
2. The proposed replicable active learner is the first of its kind. Although simple, it represents a solid first step toward tackling this problem.
3. The sample complexity bounds are clean and directly parallel the improvements achieved by active learners over passive learners in the standard PAC setting.
4. The introduction and background are well-written, providing sufficient context and a thorough overview of related work.

Weakness:
1. Many of the technical contributions appear to be adaptations of Bun et al. (2023), rather than novel developments.
2. The technical sections (particularly Sections 3 and 4) feel repetitive and overly lengthy. While the introduction and background are well-written, they could be more concise, and the main body of the paper would benefit from tighter presentation.
3. The technical exposition is largely descriptive but offers limited insight. In particular, some design choices in the algorithms raise questions:


    a. Why is it necessary to approximate the disagreement region using the rSTAT algorithm? Given access to unlimited unlabeled data, the    mass of the disagreement region can in principle be calculated exactly.



    b. Is it necessary to make all the version space V replicable? It feels like we can just use the replicable learning algorithm from Bun et al. (2023) on the last version space and we can get some theoretical guarantees.

**Additional Comments:**

I believe the paper would be strengthened by including a lower bound, as this would clarify how tight the upper bound is. For a theory-focused conference, the current contribution may feel insufficient, but given TMLR’s broader standards, it can still be considered a meaningful contribution.

**Audience:**

Yes

**Audience Explanation:**

As noted earlier, active learning is a classical area in learning theory, while replicable learning has emerged as a popular direction in recent years, with many researchers working on related topics. Prior to this work, no algorithms or label complexity bounds were known for this setting. This paper provides a valuable first step toward addressing this problem.

**Broader Impact Concerns:**

This is a theoretical work, and I do not see any potential ethical concerns arising from it.

**Claims And Evidence:**

Yes

**Claims Explanation:**

The paper provides rigorous mathematical proofs for all its claims. As an expert in active learning with knowledge of replicable learning, I find the results convincing at first glance. I reviewed the proofs and, upon a quick check, found them to be valid.

**Requested Changes:**

1. The writing has substantial room for improvement, particularly in the following ways:

    a. The main body of the paper could be considerably shortened. The core idea is simple and intuitive for readers familiar with active learning or replicable learning, and the content could likely fit within eight pages or fewer. Much of the technical detail could be moved to the appendix.

    b. The technical writing should focus more on how the techniques from Bun et al. (2023) are applied, rather than restating them in detail. Including additional examples and intuitive explanations would make the presentation more accessible.



2. The algorithm may be simplified (as related to my earlier comments in the Weaknesses section). If simplification is not feasible, it would be valuable to explain why applying the randomized threshold trick only at the final iteration is insufficient.

---

> ### Author Response · Authors · 2025-10-20
> **Addressing the main concerns- conciseness, algorithmic choices, future extensions**
>
> Conciseness and Focus
> ---------------------
> We significantly shortened the main body of the paper by moving proofs into the appendix -- keeping only lemmas, theorems and descriptions of the replicable algorithms.
>
>
> Using rSTAT for estimating DIS(V)
> ---------------------------------
> It is indeed possible to fully compute the disagreement region given unlimited unlabeled data. However, our rationale behind using rSTAT to replicably estimate the disagreement region was to have an algorithm that runs in finite time that could potentially be adapted for empirical validations.
>
> Ensuring replicability at the end instead of all rounds
> -------------------------------------------------------
> We note that neither of our algorithms maintain replicability of the version space at every iteration. Much like the algorithm of [1] for replicable learning of finite hypothesis classes, we only enforce that there will be small symmetric difference (relative to the union) between the final version spaces of two independent runs with shared randomness. Replicability is then only enforced at the final step of hypothesis selection from the set of candidate hypotheses. That said, we do use randomized thresholding at every iteration, but this is because reducing the version space at every round requires removing hypotheses with error exceeding some threshold, and replicability requires randomization of this threshold to ensure replicability even for a worst-case distribution/hypothesis class for which all hypotheses have error near the threshold.
>
> We have updated the paper to include some additional exposition of how the techniques of [1] are applied in the active learning setting.
>
> Lower-Bound
> -----------
> We agree that a sample complexity lower bound showing that our results are (close to) tight would significantly strengthen the paper. We added a paragraph in the "Conclusions and Future Work" section of the paper, discussing an approach to obtain lower bounds by transforming replicable algorithms into differentially private ones (See [1]). We did not find any such lower bounds in our literature review, so we left the problem to future work. That said, we expect our sample complexity to be nearly tight, based on lower bounds in terms of $\rho$ and $|H|$ for replicable learning in the passive learning setting as well as lower bounds in terms of $\theta$ and $\nu/\varepsilon$ for active learning without stability constraints.
>
> [1] "Stability is Stable" Bun, Gaboardi, Hopkins, Impagliazzo, Lei, Pitassi, Sivakumar, Sorrell '23

---

> > ### Comment · Reviewer_bvPC · 2025-10-30
> >
> > Thanks for your reply! I still think it would be meaningful to think about whether using rSTAT in every iteration is necessary.

---

> > > ### Author Response · Authors · 2025-11-10
> > > **Clarification of rSTAT usage**
> > >
> > > Every step in the algorithm needs to be replicable to ensure replicability of the entire algorithm. Thus, we need to ensure that the disagreement region is approximated to the same exact value in two different runs of our algorithms. We contemplated three ways of achieving this goal:
> > >
> > > 1. Use a deterministic computation.
> > > 2. ⁠Calculate the disagreement region exactly, as proposed by you earlier.
> > > 3. ⁠Using a replicable approximation which employs the rSTAT algorithm.
> > >
> > > Regarding 1: This is an approach we briefly considered for the sake of simplicity. While we prove that the disagreement region will be at least halved in each iteration, the disagreement region might, in reality, be reduced far more. Then the threshold would be chosen far too strictly, if we chose to assume a halved disagreement region, i.e., we could be discarding good or even the best hypothesis. With regards to the previous iteration, the threshold would be, say, halved, while the actual error rates could be conditioned on a much smaller probability region.
> > >
> > > Regarding 2: For a non-finite input space X this may result in infinite computation time which we want to avoid.
> > >
> > > Thus, we see 3 as our only option.
> > >
> > > Do let us know if we weren’t able to address your concern fully, or if you think that a short discussion about this algorithmic choice would be a valuable addition to our paper.

---

> > > > ### Comment · Reviewer_bvPC · 2025-11-10
> > > >
> > > > My question is that why every step in the algorithm needs to be replicable to ensure replicability. I think this might be relaxed. My intuition is that if we can ensure the last version space has decent portion of overlaps with high probability over the samples, then we can use some rounding tricks to achieve replicability.

---

### Review · Reviewer_Xj4P · 2025-10-01

**Summary Of Contributions:**

This paper extends two classical active learning algorithms, namely CAL and A^2, to replicable active learning algorithms that would yield identical models under independent runs with high probability. The authors show that, compared to replicable passive learning algorithms, replicable active learning algorithms enjoy substantial label savings, e.g., exponentially less labels in the realizable case.

**Audience:**

Yes

**Audience Explanation:**

The findings of this paper would be of interest to researchers working on active learning and replicable learning.

**Claims And Evidence:**

Yes

**Claims Explanation:**

The authors provide complete proofs for the stated theorems, which appear reasonable to me.

**Requested Changes:**

1. Most of the techniques used in the proofs appear to be borrowed from existing literature: (1) active learning for label savings, and (2) replicable learning results for passive learning. It would be helpful if the authors could highlight their main technical contributions.
2. While most existing active learning results (including CAL and A^2) are for infinite hypothesis classes, results presented in this paper are restricted to finite classes. The authors should highlight technical challenges involved and discuss potential directions to generalize these results to infinite classes.
3. Please clarify if the presented replicable results for active learning imply other related properties, such as stability or differential privacy, in the context of active learning.

---

> ### Author Response · Authors · 2025-10-20
> **Addressing the main concerns- infinite classes and relation to DP**
>
> Highlighting Technical Contributions
> ------------------------------------
> Our main technical contributions have been elaborated in the "Our Results" section and summarized in the "Conclusion and Future Work" section in our updated manuscript.
>
> Beyond finite classes
> ---------------------
> We agree that extending our results to infinite hypothesis classes would be a natural next step. Unfortunately, existing active learning upper-bounds that depend on VC dimension of the hypothesis class do not hold in our case, due to lower-bounds for private learning (and therefore replicable learning) that require finite Littlestone dimension. It would certainly be an interesting question to show that finite Littlestone dimension and therefore globally stable learning can still give the sample complexity improvements that we target in this work via active learning, but we defer this question to future work.
>
> Relation to other notions of Stability
> --------------------------------------
> The replicability$\to$DP transformation[1] applies in our active-learning setting (the reverse does not). To our knowledge, lower bounds specific to differentially private active learning are not yet established clearly; any such bounds would likely inform the tightness of our replicable active-learning guarantees via this connection.
>
>
> We have incorporated these clarifications and pointers in the revised manuscript, mostly adding to the Conclusion and Future Works section.
>
> [1] "Stability is Stable" Bun, Gaboardi, Hopkins, Impagliazzo, Lei, Pitassi, Sivakumar, Sorrell '23

---

### Review · Reviewer_8GiF · 2025-10-03

**Summary Of Contributions:**

This paper addresses replicability in active learning (CAL and A^{2}). Each algorithm combines rSTAT subroutines to estimate quantities from unlabeled samples with random thresholding to ensure that independent runs using shared randomness produce the same predictor with high probability. The authors provide sample-complexity bounds that ensure replicable recovery of the optimal hypothesis. In both cases they have a discussion to compare it with the passive version.

**Additional Comments:**

N/A

**Audience:**

Yes

**Audience Explanation:**

Yes. Replicability is a central theme in contemporary learning theory and closely connected to differential privacy and algorithmic stability. This work offers the first (to my knowledge) replicable variants of CAL/A for finite classes, with explicit label-complexity guarantees—of clear interest to (i) theory readers following stability/DP/replicability, (ii) active learning researchers concerned with disagreement-based methods, and (iii) practitioners curious about enforcing replicability without sacrificing label efficiency. While the current scope is finite classes, the techniques suggest directions for extending to infinite classes, which should further engage the TMLR audience.

**Claims And Evidence:**

Yes

**Claims Explanation:**

The core claims are formalized as Theorem 1 (realizable/RepliCAL) and Theorem 2 (agnostic/ReplicA2) and are backed by proofs that adapt standard disagreement-based analyses (CAL/A² halving of the disagreement mass) and the good/bad-threshold coupling used in replicable passive learning. The label-complexity statements track the intended dependencies on the disagreement coefficient , accuracy parameters , and the replicability parameter ρ; the sources of polylog factors (rSTAT tolerances, per-round union bounds, boosting) are identifiable in the arguments.

**Requested Changes:**

Include a table for symbols
Some formulas has number and some does not have. Please pick one style.
Extend boosting section in more detail

---

> ### Author Response · Authors · 2025-10-20
> **Addressing the main concerns- Presentation improvements and Boosting**
>
> Presentation Improvements
> --------------------------------
> We added a table of symbols in the appendix and linked to it in the main body of the paper. We also unified the equation numbering style to always include the number.
>
> Boosting Section
> --------------------
> The boosting procedure is taken from [1] without adaptation. Section 3.3 of our updated paper provides a high-level overview of the algorithm giving an intuition for the dependence of $\rho$ and $\delta$.  We avoided going further into the technical details because we do not make any additional contributions to the explanation given in the cited source.
>
> [1] "Stability is Stable" Bun, Gaboardi, Hopkins, Impagliazzo, Lei, Pitassi, Sivakumar, Sorrell '23

---

### Decision · Action_Editor_L62X · 2025-11-14

**Recommendation:** Accept as is

**Audience:**

Yes

**Audience Explanation:**

The reviewer think that the paper provides the first set of results on replicable active learning. Replicable learning is an emerging direction of good interest to some TMLR audience; it can also benefit active learning and learning theory audience.

**Claims And Evidence:**

Yes

**Claims Explanation:**

The reviewer think that the proposed method augment the active learning algorithm CAL and A^2, with randomized thresholding technique of Bun et al. They think that the bounds correctly track the dependence on the relevant quantities, and the proofs appear to be technically rigorous.

---

> ### Author Response · Authors · 2025-12-15
>
> Dear TMLR editors and reviewers,
>
> While preparing the camera-ready version of our paper in response to the reviewer comments, we identified a technical bug that requires small changes to the main algorithms in our paper and their associated proofs. We have a plan to update our proofs to fix this bug, but are still working on updating the paper. This is our first time submitting to TMLR, so I'm not sure what the process is for adding new rounds of review, but we hope to have the planned changes incorporated soon and would appreciate guidance on next steps for these updates.
>
> Thank you very much for your time and consideration! Best,
> Jess